# Constant-Modulus-Waveform Design for Multiple-Target Detection in Colocated MIMO Radar

**DOI:** 10.3390/s19184040

**Published:** 2019-09-19

**Authors:** Bingfan Liu, Baixiao Chen, Minglei Yang

**Affiliations:** National Laboratory of Radar Signal Processing, Xidian University, Xi’an 710071, Chinamlyang@xidian.edu.cn (M.Y.)

**Keywords:** MIMO radar, constant waveform design, multiple-target detection

## Abstract

For improving the performance of multiple-target detection in a colocated multiple-input multiple-output (MIMO) radar system, a constant-modulus-waveform design method is presented in this paper. The proposed method consists of two steps: simultaneous multiple-transmit-beam design and constant-modulus-waveform design. In the first step, each transmit beam is controlled by an ideal orthogonal waveform and a weight vector. We optimized the weight vectors to maximize the detection probabilities of all targets or minimize the transmit power for the purpose of low intercept probability in the case of predefined worst detection probabilities. Various targets’ radar cross-section (RCS) fluctuation models were also considered in two optimization problems. Then, the optimal weight vectors multiplied by ideal orthogonal waveforms were a set of transmitted waveforms. However, those transmitted waveforms were not constant-modulus waveforms. In the second step, the transmitted waveforms obtained in the first step were mapped to constant-modulus waveforms by cyclic algorithm. Numerical examples are provided to show that the proposed constant-waveform design method could effectively achieve the desired transmit-beam pattern, and that the transmit-beam pattern could be adaptively adjusted according to prior information.

## 1. Introduction

Multiple-input and multiple-output (MIMO) radars have become the focus of intensive research in recent years [1,2,3]. A MIMO radar is generally defined as a radar system with multiple-transmit linearly independent waveforms that enables the joint processing of data received by multiple receiving antennas. MIMO radar antennas may be widely separated [2] to improve target-detection capabilities or colocated to improve spatial resolution, parameter identifiability, and interference-rejection capabilities [3].

A colocated MIMO radar, in which transmit and receive antennas are closely spaced, is similar to the phased-array radar, except that the transmitted waveforms are correlated or uncorrelated with each other [4,5,6]. In other words, a phased array can be regarded as a special case of a colocated MIMO radar. A flexible transmit beam pattern is one of the most significant advantages of the colocated MIMO radar, because a MIMO radar has many more degrees of freedom (DOF) than a phase-array radar, and it can achieve a beam pattern significantly closer to the desired beam pattern [7]. The conventional approach for a transmit beam pattern includes two steps [6,7]. The first step is to optimize covariance matrix R of transmitted waveforms under a total power constraint or an element power constraint to achieve the desired beam pattern [6,7] or to satisfy the minimum parameter-estimation error [8]. The second step is to determine the waveform matrix whose covariance matrix is equal or close to R [6]. In this step, an efficient algorithm, the cyclic algorithm (CA), is proposed to obtain the constant modulus or low peak-to-average-power ratio (PAR) waveforms with good auto- and cross-correlation properties.

However, with the increase of the number of antennas, the optimal transmitted waveforms or covariance matrix of the transmitted waveforms cannot be figured out immediately due to heavy computation. A more efficient approach was proposed to synthesize transmitted signals by designing a weight matrix given a set of orthogonal waveforms [9,10,11]. By optimizing the weight matrix, the paper [9] focused the transmitting energy within a certain spatial sector to improve the Signal-to-Noise Ratio (SNR) gain. Using a similar method, a study [10,11] formed the desired beam patterns with lower sidelobe levels for generalized MIMO configurations. Then, the waveform-design problem was converted into the optimization of the weight matrix with the assumption of complete orthogonal waveforms. The transmitted waveform could aso be expressed by a weighted sum of orthogonal waveforms. However, the approach could not ensure that the transmitted waveforms satisfied the constant-modulus constraint.

Desired transmit-beam patterns generally fall into two categories: broad beams for search mode and multiple pencil beams for the track mode of multiple targets [7]. It is worth noting that both categories of beam shapes can be realized by the method in [4,5,6,7]. In fact, this method could design beams of any shape, of course, including wide beams and multiple beams. A novel range-angle- dependent beam pattern was formed in a frequency-diverse-array MIMO (FDA-MIMO) radar [12,13]. Based on the MIMO radar, the FDA-MIMO radar is capable of employing a small frequency increment across the array elements and utilize degrees of freedom in the range–angle domains to jointly determine the range and angle parameters of the target.

Here, we mainly discuss multiple beams for target tracking. At locations like airports, parking lots, and intersections, the angles and distances of the targets with respect to the radar greatly vary, which requires the radar to have flexible and fast beam-forming capabilities. The colocated MIMO radar for multiple-target tracking has been investigated in recent years [14,15,16,17,18,19,20,21,22] due to a flexible transmit beam pattern. In [14,15], a simultaneous multibeam resource-allocation scheme was developed to improve the worst tracking performance by adjusting the number, direction, and power of the beams. As its extension, a study [17] developed joint beam selection and a power-allocation strategy for a netted colocated MIMO radar system. In another study [18], the probabilistic uncertainty on the target radar cross-section parameter was taken into account in a colocated MIMO radar system, and the total power consumption of multiple beams was minimized to meet a specified multiple-target localization-accuracy requirement with high probability. However, the focus in [14,15,17,18] is how to design the desired multibeam pattern for certain purposes instead of how to synthesize the desired multibeam pattern with the designed transmitted waveforms or the covariance matrix of the transmitted waveform.

In [19], the waveform was determined to minimize the Bayesian Cramér–Rao bound (BCRB) or the Reuven–Messer bound (RMB) for estimation of unknown system parameters. The corresponding transmit beam was automatically focused on targets. This method was also applied to a target-tracking problem in which the transmit beam pattern was sequentially determined based on historical observations [20]. Another similar approach of designing waveforms is to minimize the expected mean squared error [21]. However, the computational load in [19,20,21] was heavy because of the heavy computation of the posterior probability density function of target parameters. Based on prior information, a robust waveform design was proposed to maximize the detection probability of multiple targets [23].

In [6,7,8,19,20,23], a transmit beam pattern was obtained by optimizing the covariance matrix of the transmitted waveforms, which is a Semidefinite Programming (SDP) problem. Simultaneous multiple pencil beams could actually be determined by beam direction and amplitude. If the beam direction and amplitude, instead of the covariance matrix of transmitted waveforms, are optimized in designing simultaneous multiple pencil beams, the dimension of the optimization variable is greatly reduced. Based on this analysis, we propose a novel and concise signal model where each beam is controlled by an ideal orthogonal waveform and a weight vector. For improving the performance of multiple-target detection, we optimized the weight vectors to maximize the detection probabilities of all targets. When transmit power was enough to achieve good detection performance, we minimized the transmit power for the purposes of low intercept probability in the case of predefined worst detection probabilities. The ideal transmitted waveforms were expressed by the product of ideal orthogonal waveforms and the optimal weight vectors. However, these ideal transmitted waveforms were not constant-modulus waveforms. Inspired by work in [6], we chose a set of constant-modulus waveforms to approximate the ideal transmitted waveforms by a cyclic algorithm. Furthermore, numerical examples are provided to show that the proposed constant-waveform design method could effectively achieve the desired transmit beam pattern, and that the transmit beam pattern could be adaptively adjusted according to prior information.

The rest of this paper is organized as follows. Section 2 presents the proposed signal model for the MIMO radar. In Section 3, two optimization problems for simultaneous multiple transmit-beam design are introduced. In Section 4, the development of the cyclic algorithm for constant waveform design is outlined and detailed. Simulation results are provided in Section 5, and this paper is concluded in Section 4.

*Notation*: Boldface uppercase letters stand for matrices. Boldface lowercase letters stand for column vectors. ${\left(\cdot \right)}^{*}$, ${\left(\cdot \right)}^{T}$, and ${\left(\cdot \right)}^{H}$ denote the conjugate, transpose, and conjugate transpose, respectively; ⊗, Kronecker product; ⊛, convolution operation; ⊙, Hadamard product; |·|, modulus of complex scalar or element-wise modulus of a complex vector; ∥·∥, $\ell_{2}$-norm; ${∥\cdot ∥}_{F}$, Frobenius norm; ${\left(\cdot \right)}^{-\frac{1}{2}}$, inverse of Hermitian square root of a matrix; and tr{·} and Re{·}, trace and real part, respectively.

## 2. Signal Model

Consider a colocated MIMO radar equipped with *M* transmit antennas and *N* receive antennas. Both the receive and transmit arrays are uniform linear arrays with half-wavelength spacing between the adjoining antennas. Assuming that *K* point targets of interest are located at ${\left\{\theta_{k}\right\}}_{k=1}^{K}$, we next correspondingly generated *K* transmit simultaneous beams. We also assumed that the *k*th beam emitted *k*th narrow band waveform $\phi_{k}$, and that waveform matrix $\phi \left(t\right)={\left[\phi_{1}\left(t\right),\phi_{2}\left(t\right),\cdots ,\phi_{K}\left(t\right)\right]}^{T}$ satisfied the following orthogonal condition:(1)$$\int_{T_{p}}\phi (t)\phi^{H}\left(t\right)dt=I_{K}$$ where *t* refers to the time index within the radar pulse, $T_{p}$ is radar pulse width, and $I_{K}$ is a K×K identity matrix.

Then, we focused on how to simultaneously generate *K* transmit beams. We prepared *K* weight vectors for *K* beams or targets, and the *k*th weight can be given simply by:(2)$$w_{k}=\sqrt{\alpha_{k}}\cdot a_{t}^{*}\left(\theta_{k}\right)$$ where $\alpha_{k}$ ($\sum_{k=1}^{K}\alpha_{k}=1,\alpha_{k}>0$) is the normalization factor of the assigned power of the *k*th beam, $\theta_{k}$ is the direction of the *k*th beam, and $a_{t}\left(\theta \right)$ is the transmit steering vector. Therefore, all signals radiating toward a hypothetical target located at direction θ could be constructed as
(3)$$s\left(\theta ,t\right)={\left(a_{t}^{T}\left(\theta \right)W\phi \left(t\right)\right)}^{T}$$ where $W=[w_{1},w_{2},\cdots ,w_{K}]$ is the weight matrix. The waveform transmitted from the antennas can be expressed as x(t)=Wϕ(t), which should satisfy practical constraints (such as the constant modulus) so that nonlinear amplifiers can be used. Note that orthogonal waveform vector ϕ(t) here is just an ideal hypothetical auxiliary variable, and our ultimate goal was to design transmitted waveform x(t).

Then, the transmit beam pattern, i.e., the distribution of transmitted signal energy in space, is given by [4,5]    

(4)$$\begin{array}{cc} S(\theta ) & =\int_{T_{p}}s^{T}\left(\theta ,t\right)s^{*}\left(\theta ,t\right)dt\\ \; & =a_{t}^{T}\left(\theta \right)W\cdot \int_{T_{p}}\phi (t)\phi^{H}\left(t\right)dt\cdot W^{H}a_{t}^{*}\left(\theta \right)\\ \; & ={∥a_{t}^{T}\left(\theta \right)W∥}^{2}\\ \; & =\sum_{k=1}^{K}{\left|a_{t}^{T}\left(\theta \right)w_{k}\right|}^{2} \end{array}$$

Note that transmit beam pattern S(θ) is equal to the sum of the *K* beam patterns.

The baseband representation of the received signal can be modeled as:(5)$$\begin{array}{c} y\left(t\right)=\sum_{k=1}^{K}r_{k}\sqrt{h_{k}E}\;a_{r}\left(\theta_{k}\right)a_{t}^{T}\left(\theta_{k}\right)x(t-\tau_{k})+n\left(t\right) \end{array}$$ where $r_{k}$ is the reflection complex coefficients of the *k*th target, $h_{k}\propto 1/R_{k}^{4}$ is the variation in signal strength due to path-loss effects ($R_{k}$ is range between *k*th target and radar), *E* denotes the total power of the radar, $a_{r}\left(\theta \right)$ is the receive steering vector, $\tau_{k}=2R_{k}/c$ (*c* denotes speed of light) is time delay, and n(t) denotes the white Gaussian noise of the sensor array. Using matched filter $x^{*}\left(-t\right)$, we can obtain an N×M matrix:(6)$$\begin{array}{cc} Z(t) & =y\left(t\right)⊛x^{H}\left(-t\right)\\ \; & =\int_{T_{p}}y(u)x^{H}\left(u-t\right)du\\ \; & =\sum_{k=1}^{K}r_{k}\sqrt{h_{k}E}\;a_{r}\left(\theta_{k}\right)a_{t}^{T}\left(\theta_{k}\right)\\ \; & \;\int_{T_{p}}x(u-\tau_{k})x^{H}\left(u-t\right)du+N\left(t\right). \end{array}$$

Note that matched filter $x^{*}\left(-t\right)$ is actually a set of matched filters $[{x_{1}}^{*}\left(-t\right),{x_{2}}^{*}\left(-t\right),\cdots ,$${x_{M}}^{*}{\left(-t\right)]}^{T}$. The N×M receive matrix in the range cell of the *k*th target ($t=\tau_{k}$) can be written as:(7)$$\begin{array}{cc} V_{k} & =r_{k}\sqrt{h_{k}E}\;a_{r}\left(\theta_{k}\right)a_{t}^{T}\left(\theta_{k}\right)\\ \; & \;\int_{T_{p}}x(u-\tau_{k})x^{H}\left(u-\tau_{k}\right)du+\tilde{N} \end{array}$$ where $\tilde{N}\in C^{N\times M}$ is narrow band Gaussian noise. Then, analog signal x(t) is discretized as $X\in C^{M\times L}$, where *L* is the length of the discrete signal. Ideal orthogonal waveform ϕ(t) can be discretized as $\Phi \in C^{K\times L}$, which satisfies

(8)$$\begin{array}{c} \frac{1}{L}\Phi \Phi^{H}=I_{K}. \end{array}$$

We then have

(9)X=WΦ.

The integral in Equation (Equation 7) can be turned into sums:(10)$$\begin{array}{cc} \; & \int_{T_{p}}x(u-\tau_{k})x^{H}\left(u-\tau_{k}\right)du\\ ⟹ & X\cdot \left\{X^{H}{\left(XX^{H}\right)}^{-\frac{1}{2}}\right\}={\left(XX^{H}\right)}^{\frac{1}{2}} \end{array}$$

Here, we used normalized matched filter $X^{H}{\left(XX^{H}\right)}^{-\frac{1}{2}}$ instead of $X^{H}$ for range compression [8].

Then, Equation (Equation 7) can be vectorized as:
(11)$$\begin{array}{cc} v_{k} & =r_{k}\sqrt{h_{k}E}\;a_{r}\left(\theta_{k}\right)⊗\left({\left(XX^{H}\right)}^{\frac{1}{2}}a_{t}\left(\theta_{k}\right)\right)+n\\ \; & =r_{k}\sqrt{h_{k}E}\;c\left(\theta_{k}\right)+n \end{array}$$ where MN×1 vector
(12)$$\begin{array}{c} c\left(\theta \right)=a_{r}\left(\theta \right)⊗\left({\left(XX^{H}\right)}^{\frac{1}{2}}a_{t}\left(\theta \right)\right) \end{array}$$ is the virtual steering vector, and n is the MN×1 noise term whose covariance is given by $R_{n}=\sigma^{2}I_{MN}$, where $\sigma^{2}$ is the noise power. The covariance matrix of the noise is detailed in [8] (Equation (Equation 7)).

## 3. Simultaneous Multiple Transmit-Beam Design

In a MIMO radar, we can dynamically design transmitted beams with prior target information. Our aim was to ensure that each target could be covered by enough energy and be detected with high detection probability. In this paper, we took false-alarm probability $P_{\mathrm{FA}}$ and detection probability $P_{D}$ as the evaluation criterion of the transmitted beam design. It is well known that the SNR critically influences $P_{\mathrm{FA}}$ and $P_{D}$. Therefore, in this section, the target SNR was first investigated. Then, we propose two optimization problems to realize the transmitted beam design.

### 3.1. SNR Analysis

For the *k*th target, we used conventional nonadaptive beam former $w_{d}=c\left(\theta_{k}\right)$ and obtained the corresponding SNR:(13)$$\begin{array}{cc} SNR_{k} & \overset{△}{=}\frac{{\left|w_{d}^{H}c\left(\theta_{k}\right)\right|}^{2}}{w_{d}^{H}R_{n}w_{d}}\\ \; & =\gamma_{k}^{2}Eh_{k}\frac{c^{H}\left(\theta_{k}\right)c\left(\theta_{k}\right)}{c^{H}\left(\theta_{k}\right)R_{n}c\left(\theta_{k}\right)}\\ \; & =\frac{\gamma_{k}^{2}Eh_{k}}{\sigma^{2}}\frac{{\left|a_{r}^{H}\left(\theta_{k}\right)a_{r}\left(\theta_{k}\right)\right|}^{2}{\left|a_{t}^{H}\left(\theta_{k}\right)\left(XX^{H}\right)a_{t}\left(\theta_{k}\right)\right|}^{2}}{\left|a_{r}^{H}\left(\theta_{k}\right)a_{r}\left(\theta_{k}\right)\right|\left|a_{t}^{H}\left(\theta_{k}\right)\left(XX^{H}\right)a_{t}\left(\theta_{k}\right)\right|}\\ \; & =\frac{\gamma_{k}^{2}}{\sigma^{2}}Eh_{k}NL\left|a_{t}^{H}\left(\theta_{k}\right)\left(XX^{H}\right)a_{t}\left(\theta_{k}\right)\right| \end{array}$$ where $\gamma_{k}^{2}=E\left\{r_{k}^{2}\right\}$ is the variance of the *k*th target refection coefficient.

A conventional MIMO radar transmits orthogonal waveforms, i.e., $XX^{H}=I$. In such cases, output SNR can be rewritten as:(14)$$\begin{array}{c} SNR_{O-\mathrm{MIMO}k}=\frac{\gamma_{k}^{2}}{\sigma^{2}}Eh_{k}NML. \end{array}$$

In this paper, the transmitted waveform was correlated, whose covariance matrix is described as:(15)$$\begin{array}{cc} XX^{H} & =W\Phi \Phi^{H}W^{H}\\ \; & =L\cdot WW^{H}. \end{array}$$

Substituting Equation (Equation 15) in Equation (Equation 13), the corresponding output SNR can be expressed as:(16)$$\begin{array}{cc} SNR_{C-\mathrm{MIMO}k} & =\frac{\gamma^{2}}{\sigma^{2}}Eh_{k}NL\left|a_{t}^{H}\left(\theta_{k}\right)\left(WW^{H}\right)a_{t}\left(\theta_{k}\right)\right|\\ \; & =\frac{\gamma^{2}}{\sigma^{2}}Eh_{k}NL\sum_{i=1}^{K}{\left|a_{t}^{T}\left(\theta_{k}\right)w_{i}\right|}^{2}\\ \; & ≃\frac{\gamma^{2}}{\sigma^{2}}Eh_{k}NM^{2}L\alpha_{k} \end{array}$$

In Formula (Equation 16), we assumed that targets were widely separated. Thus, $\sum_{i=1}^{K}{\left|a_{t}^{T}\left(\theta_{k}\right)w_{i}\right|}^{2}$ was approximately equal to ${\left|a_{t}^{T}\left(\theta_{k}\right)w_{k}\right|}^{2}$. By comparing Equations (Equation 14) and (Equation 16), it could be found that the MIMO radar with a correlated waveform in this paper focused energy on targets, while the MIMO radar with a uniform transmit beam pattern spreads energy evenly throughout the space. From Formula (Equation 16), the SNRs of targets could be adjusted by changing the normalization factor of assigned power $\alpha_{k}$.

Consider a special case in which only one beam (K=1) points to the only target. Then, the MIMO system is simplified to a phased array radar that maximizes the SNR at a given focal point over the class of possible active arrays [24].

### 3.2. Transmit-Beam Design Based on Robust Energy Allocation

In radar detection, detection probability $P_{D}$ can be expressed as a function of several related variables [25]:(17)$$P_{D}=f\left(P_{\mathrm{FA}},SNR,N_{p},type,NC\right)$$ where $N_{p}$ denotes integration number, type denotes the various target RCS fluctuation model, and NC denotes the noncoherent integration or coherent integration.

Our goal here was to design W so that each target could be detected with the greatest probability. We maximized the worst-case $P_{D}$ of targets, and the corresponding optimization problem could be formulated as:(18)$$\begin{array}{cc} \; & \max_{W}\;\min_{k=1,\cdots ,K}P_{D}\left(k\right) \end{array}$$

Due to the structure of W in (Equation 2), optimization variables could be replaced by ${\left\{\alpha_{k}\right\}}_{k=1}^{K}$ when prior information is known. Substituting Equation (Equation 16) into Equation (Equation 17), the objective function of Equation (Equation 18) can be rewritten as:(19)$$P_{D}\left(k\right)=f_{k}\left(\alpha_{k}\right)$$ where $P_{\mathrm{FA}}$, $N_{p}$, NC are predefined, and $f_{k}\left(\alpha_{k}\right)$ denotes the nonlinear function for calculating the probability of detection of the *k*th target [25]. Thus, optimization Problem (Equation 18) can be reformulated as:(20)$$\begin{array}{cc} \; & \min_{\alpha_{k}\left(k=1,\cdots ,K\right)}\;-t\\ \; & s.t.\;f_{k}\left(\alpha_{k}\right)\geq t\;\left(k=1,\cdots ,K\right)\\ \; & \;\sum_{k=1}^{K}\alpha_{k}=1. \end{array}$$

Note that Equation (Equation 20) is nonlinear constraint optimization problem due to nonlinearity of function (Equation 19). Usually, it is difficult to take derivative of $P_{D}$ with respect to SNR or $\alpha_{k}$. Therefore, we could not directly obtain the solution of Problem (Equation 20).

However, we could see that the essence of optimization Problem (Equation 20) was that each target obtained equal and maximum probability of detection. For solving Problem (Equation 20), we defined:(21)$$\begin{array}{cc} \; & \alpha_{k}=h_{k}\left(P_{d}\right)=f_{k}^{\left(-1\right)}\left(P_{d}\right) \end{array}$$(22)$$\begin{array}{cc} \; & \alpha =g\left(P_{d}\right)=\sum_{k=1}^{K}\alpha_{k}=\sum_{k=1}^{K}h_{k}\left(P_{d}\right) \end{array}$$ where $f_{k}^{\left(-1\right)}$ is the inverse function of $f_{k}$, and $P_{d}$ is the common probability of detection of *K* targets. Because $P_{d}$ increases with the increasing $SNR_{k}$ (or $\alpha_{k}$) [25], $f_{k}\left(\alpha_{k}\right)$, $h_{k}\left(P_{d}\right)$, and $g(P_{d})$ were all monotone increasing functions. Therefore, optimization Problem (Equation 20) could be equivalent to solve the following equations:    

(23)$$\begin{array}{cc} \; & g(P_{d})=1 \end{array}$$

(24)$$\begin{array}{cc} \; & \alpha_{k}=h_{k}\left(P_{d}\right)\;\left(k=1,\cdots ,K\right) \end{array}$$

Benefitting from the monotonicity of $g(P_{d})$, we propose a binary-search algorithm that can be expressed as Algorithm 1.

**Algorithm 1** Binary-search algorithm for Pd**Initialization:** Find arbitrary two values Pdl and Pdr that satisfy g(Pdl)<1 and g(Pdr)>1. Let Pdm=(Pdl+Pdr)/2, μ=10−4.**Output:**
Pdm **while**
|g(Pdm)−1|>μ
**do**  **if**
$g(P_{\mathrm{dm}})>1$
**then**   $P_{\mathrm{dr}}=P_{\mathrm{dm}}$.  **else**   $P_{\mathrm{dl}}=P_{\mathrm{dm}}$.  **end if**  $P_{\mathrm{dm}}=\left(P_{\mathrm{dl}}+P_{\mathrm{dr}}\right)/2$
 **end while**

In Algorithm 1, the evaluation of function $g(P_{d})$ can be divided into *K* parts, as described in Equation (Equation 21). For each evaluation part, it was difficult to describe function $h_{k}\left(P_{d}\right)$ with a specific form. For a given $P_{d}$, it is difficult to obtain $\alpha_{k}$. However, given $\alpha_{k}$, it is easy to calculate $P_{d}$ by function $f_{k}\left(\alpha_{k}\right)$ (Equation (Equation 19)). Therefore, we could also calculate $h_{k}\left(P_{d}\right)$ through the binary-search algorithm, which is described in Algorithm 2.

**Algorithm 2** Binary-search algorithm for αk**Initialization:** Given Pd. Find arbitrary two value αkl and αkr, which satisfy fk(αkl)<Pd and fk(αkr)>Pd. And Let αkm=(αkl+αkr)/2, μ=10−4.**Output:**
αkm **while**
|fk(αkm)−Pd|>μ
**do**  **if**
$f_{k}\left(\alpha_{km}\right)>P_{d}$
**then**   $\alpha_{kr}=\alpha_{km}$.  **else**   $\alpha_{kl}=\alpha_{km}$.  **end if**  $\alpha_{km}=\left(\alpha_{kr}+\alpha_{kl}\right)/2$ **end while**

If all *K* targets belong to the same Swerling target, the detection probabilities of targets under the same SNR are identical. We could replace detection probability $P_{D}$ by the SNR in the optimization problem. Corresponding optimization Problem (Equation 20) could be simplified as:(25)$$\begin{array}{cc} \; & \min_{\alpha_{k}\left(k=1,\cdots ,K\right)}\;-t\\ \; & s.t.\;SNR_{k}\left(\alpha_{k}\right)\geq t\;\left(k=1,\cdots ,K\right)\\ \; & \;\sum_{k=1}^{K}\alpha_{k}=1. \end{array}$$

Problem (Equation 25) is clearly a linear-programming (LP) problem, and corresponding Solution $\alpha_{k}\left(k=1,\cdots ,K\right)$ could be directly obtained: (26)$$\begin{array}{c} \alpha_{k}=\frac{\xi_{k}}{\sum_{k=1}^{K}\xi_{k}}\;\left(k=1,\cdots ,K\right) \end{array}$$ where $\xi_{k}=1/\left(\gamma_{k}^{2}Eh_{k}NM^{2}L/\sigma^{2}\right)$.

### 3.3. Transmit-Beam Design Based on Minimum-Energy Allocation

Sometimes, for the purpose of low intercept probability, we aim for the radar to transmit as little energy as possible in case that the probability of target detection meets predefined probabilities. Consequently, a mathematical formulation of the problem can be written as follows:(27)$$\begin{array}{cc} \; & \min_{\alpha_{k}}\sum_{k=1}^{K}\alpha_{k}\\ \; & s.t.\;f_{k}\left(\alpha_{k}\right)\geq t_{k}\;\left(k=1,\cdots ,K\right) \end{array}$$ where $t_{k}$ is the predefined detection probability of the *k*th target.

We converted optimization Problem (Equation 27) into *K* minimization problems:(28)$$\begin{array}{c} \left\{\begin{array}{c} \min\;\alpha_{k}\\ s.t.\;f_{k}\left(\alpha_{k}\right)\geq t_{k}\; \end{array}\right.\left(k=1,\cdots ,K\right). \end{array}$$

Because $f_{k}\left(\alpha_{k}\right)$ is a monotone increasing function, the solution of the *k*th minimization problem can be given directly as:(29)$$\alpha_{k}=f_{k}^{\left(-1\right)}\left(t_{K}\right).$$

Again, we used Algorithm 2 to calculate $f_{k}^{\left(-1\right)}\left(t_{K}\right)$.

### 3.4. Constant-Waveform Design

With W or ${\left\{\alpha_{k}\right\}}_{k=1}^{K}$ obtained in the previous optimization problem, our goal was to find a constant-modulus waveform matrix X, which satisfies condition X=WΦ. However, equation X=WΦ is usually inconsistent. Thus, we obtained a closed-form solution by solving the corresponding least-squares problem:(30)$$\begin{array}{cc} \; & \min_{X,\Phi ,\eta }{∥W\Phi -\eta X∥}_{F}^{2}\\ \; & s.t.\;|x_{m,l}|=1,\;\left(m=1,\cdots ,M\;l=1,\cdots ,L\right)\\ \; & \;\;\eta >0\\ \; & \;\frac{1}{L}\Phi \Phi^{H}=I_{K} \end{array}$$ where $x_{m,l}$ is the (m,l)th element of X, η is a scaling factor, and the waveform modulus was set to 1.

Note that there are three variables in the nonconvex minimization Problem (Equation 30). Inspired by [6], we used a cyclic (iterative) minimization algorithm for solving Problem (Equation 30).

For given X and η, the object function in Problem (Equation 30) could be rewritten as:(31)∥WΦ−ηX∥F2=tr(WΦΦHWH)+η2tr(XXH)−2η·Re{tr(ΦHWHX)}=L·tr(WWH)+η2tr(XXH)−2η·Re{tr(ΦHWHX)}=const−2η·Re{tr(ΦHWHX)}.

Let
(32)$$\begin{array}{c} W^{H}X=\bar{U}\Lambda {\tilde{U}}^{H} \end{array}$$ denote singular value decomposition, where $\bar{U}$ is a K×K unitary matrix, $\tilde{U}$ is a L×K semiunitary matrix, and Λ is a K×K diagonal matrix. We minimized Problem (Equation 31), and solution Φ could be expressed as [7]:(33)$$\begin{array}{c} \Phi =\bar{U}{\tilde{U}}^{H} \end{array}$$

Let $\hat{X}=W\Phi$. For a given Φ, the solution of Problem (Equation 30) under the constant-modulus criterion could be directly given by: (34)$$\begin{array}{c} x_{m,l}=\exp\left\{\arg\left({\hat{x}}_{m,l}\right)\right\} \end{array}$$(35)η=Re{tr(XHX^)}tr(XHX) where ${\hat{x}}_{m,l}$ is the (m,l)th element of $\hat{X}$.

We summarized the steps of the cyclic minimization algorithm as follows:**Step 0:** Obtain desired matrix W in the previous optimization problem. Element $x_{m,l}$ in X was set to $\left\{\exp\left(j2\pi \psi_{m,l}\right)\right\}$, where $\left\{\psi_{m,l}\right\}$ were independent random variables uniformly distributed in [0,2π].**Step 1:** Obtain Φ according to Equation (Equation 33).**Step 2:** Obtain X and η according to Equations (Equation 34) and (35).**Iteration:** Repeat Steps 1 and 2 until prespecified stop criterion is satisfied, e.g., ${∥\Phi \left(i+1\right)-\Phi \left(i\right)∥}_{F}^{2}\leq \mu$ , where Φ(i) denotes the orthogonal waveform matrix at *i*th iteration, and μ is a predefined threshold.

It is worth noting that both Equations (Equation 33) and (Equation 34) are irrelevant to scale factor η. Thus, it was not necessary to calculate the η during the iterative process.

## 4. Simulation and Discussion

To illustrate the effectiveness of the proposed method, we present simulation results in this section. Throughout our simulations, we assumed that the transmitter and receiver shared uniform linearity of M=N=32 antennas with half-wavelength interelement spacing, waveform length *L* was 128, total transmit power was set as 10 kilowatts, the power of noise $\sigma^{2}$ was set as $10^{-8}$ watts, false-alarm probability $P_{\mathrm{FA}}$ was set as $10^{-8}$, and the number of samples noncoherently integrated $N_{p}$ was set as 10. In the simulation, it was assumed that there were three targets, whose tracks with respect to the radar system are shown in Figure 1. We chose 10 moments during tracking to analyze two proposed methods of beam design. The time interval between two moments was 10 s, and tracking duration was 100 s. For convenience, the RCSs of three targets were set as $6\;m^{2}$.

### 4.1. Transmit-Beam Design Based on Robust Energy Allocation

In this case, we assumed that the RCS fluctuation models of three targets were Swerling 1, 2, and 3. Figure 2 shows a plot of the corresponding detection probabilities as a function of SNR for $N_{p}=10$ and $P_{\mathrm{FA}}=10^{-8}$.

We solved optimization Problem (Equation 20) (optimization was based on prior information), and solutions in 10 moments are shown in Figure 3.

Figure 4 compares the single-pulse SNR of three targets at various time points. According to Equation (Equation 17), the corresponding probabilities of detection are given in Figure 5. In the initial state, the single-pulse SNR of the nearest target (i.e., Target 1) was highest. Thus, the lowest energy was assigned to Target 1 (Target 1: 0.1, Target 2: 0.51, Target 3: 0.39). Over time, Targets 2 and 3 came closer to the radar system, and the distance between Target 1 and radar was nearly unchanged. In the case of constant transmit power, the single-pulse SNRs of three targets gradually increased. From Figure 2, we can see that Target 2 (Swerling 2) could achieve a given probability detection at the lowest SNR, and that Target 1 required the highest SNR for a given $P_{d}$. Therefore, in the end, the farthest target (Target 3) did not need the highest energy, while Target 1 (Swerling 1) was assigned the highest energy, as shown in Figure 3.

Consequently, Figure 5 shows the designed detection probabilities of the three targets. Note that all targets had the same detection probabilities, which was in good agreement with previous theoretical analysis. After obtaining allocation coefficient ${\left\{\alpha_{k}\right\}}_{k=1}^{K}$, the constant-modulus waveforms could be designed through the cyclic algorithm. Figure 6 shows the beam patterns in different times whose peak amplitudes were consistent with ${\left\{\alpha_{k}\right\}}_{k=1}^{K}$.

### 4.2. Transmit-Beam Design Based on Minimum-Energy Allocation

When the power of the radar system was enough to detect targets with a high detection probability, we minimized the total transmit power to reduce the probability of being intercepted by a hostile radar system.

In this case, we assumed that three targets were all Swerling 2 targets, whose detection performance was the same as Target 2 in Figure 2. Figure 7 shows the results of energy allocation in the constraint in which predefined detection probabilities were all set to 0.9. We can see that α, the sum of energy-allocation coefficients, decreased from 1 to 0.2. In the end, total transmit power was one-fifth of what is was.

Figure 8 shows comparative $P_{d}$ for the three targets with the four methods. It can be seen that our proposed method, the robust energy-allocation method described in Equation (Equation 25), performed as well as the method in [23]. For another proposed method, the minimum-energy-allocation method described in Equation (Equation 28), the detection probabilities of three targets were maintained at 0.9. Because transmitted energy with the conventional orthogonal waveform (OW) design method is distributed uniformly in space, most energy is not focused on the targets. Thus, detection probabilities using conventional orthogonal waveforms are the lowest. Figure 9 shows the transmit power of the four methods. It can be seen from this figure that the robust energy-allocation method, the method in [23], and the conventional orthogonal waveforms design method had full transmit power, while the minimum-energy-allocation method had decreasing transmit power over time.

### 4.3. Beam-Pattern Comparison

Figure 10 shows four beam patterns with different methods. To implement the method in [23] in a manner fit for comparison, we supplied the exact same prior information instead of statistical prior information. In this example, we assumed that three Swerling 1 targets were located at $\left(9\;\mathrm{km},-50^{\circ }\right)$, $\left(10\;\mathrm{km},0^{\circ }\right)$ and $\left(8\;\mathrm{km},20^{\circ }\right)$, respectively. Other parameters were consistent with the previous parameters. The beam pattern for the proposed method (ideal) was generated by weight matrix W, while the beam pattern for the proposed method (real) was generated by constant waveforms X. Except for the omnidirectional beam pattern for orthogonal waveforms, the others all exactly pointed to the targets. It was verified that the transmit beam pattern with designed constant waveforms could approximate the desired transmit beam pattern.

Figure 11 shows the beam patterns achieved by the constant-modulus waveforms of the first 10 iterations in the cyclic algorithm. The dotted line represents the ideal transmit beam pattern. Except for the beam pattern achieved by the randomly initialized waveforms being omnidirectional, the beam patterns after the first iteration all pointed to the targets well. As the number of iterations increased, the sidelobe of the beam pattern also further decreased. After 10 iterations, the beam pattern was already a good approximation of the ideal beam pattern.

Compared with the method in [23], the proposed method has two advantages: (1) low computational complexity and (2) constant waveforms. With regard to the former, weight matrix W or energy coefficients ${\left\{\alpha_{k}\right\}}_{k=1}^{K}$ could easily be obtained by a binary-search algorithm while the covariance matrix of the transmitted waveforms was obtained by solving an SDP problem. Under the assumption that all targets were the same kind of Swerling target (various target RCS fluctuation models were not considered in [23]), ${\left\{\alpha_{k}\right\}}_{k=1}^{K}$ could be obtained by solving an LP problem. For another advantage, the weight matrix in [23] could not ensure that the waveform transmitted by each antenna was constant, which prevents nonlinear amplifiers from being used in the radar system.

## 5. Conclusions

We propose a constant-modulus waveform design method especially for multiple-target detection in MIMO radars. First, we presented a novel ideal signal model in which the beam pattern for each target was controlled by a weight vector and an orthogonal waveform. Based on this ideal signal model, two energy-allocation methods were proposed for two practical scenes (constant transmit power and minimum power). For practical purposes, we used a cyclic algorithm to obtain the constant-modulus waveform, which was the approximate form of the ideal transmitted waveform (by multiplying the ideal orthogonal waveforms by the ideal weight matrix). Finally, numerical examples indicated that the proposed method could achieve better performance than the method in [23]. Furthermore, an easier optimization problem and constant waveforms help the proposed method be more practical in engineering.

However, exact prior information was considered in this paper, which is helpful to analyze the problem of waveform design, but is too ideal. Thus, future work will focus on investigating the influence of the parameter-estimation error, and finding a parameter-estimation method with historical information.

## Figures and Tables

**Figure 1 sensors-19-04040-f001:**
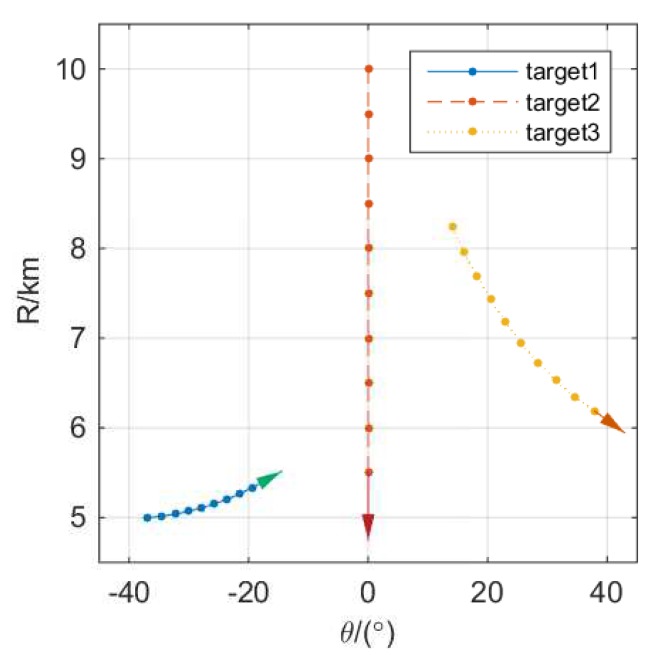
Paths of targets with Radar Cross-Sections $\left(\mathrm{RCSs}\right)=\left\{6m^{2},6m^{2},6m^{2}\right\}$. Time interval between two adjacent moments was 10 s.

**Figure 2 sensors-19-04040-f002:**
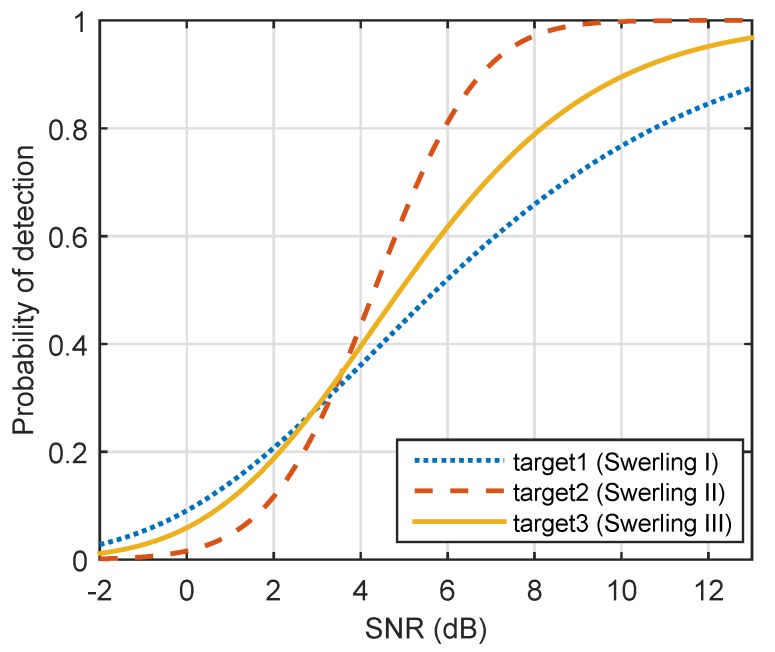
Probability of detection versus Signal-to-Noise Ratio (SNR) using noncoherent integration of 10 pulses ($N_{p}=10$) and false-alarm probability $P_{\mathrm{FA}}=10^{-8}$.

**Figure 3 sensors-19-04040-f003:**
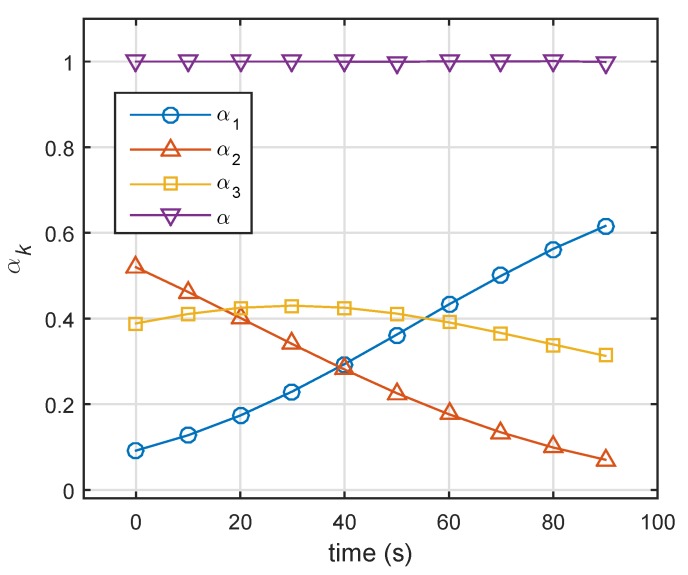
Energy-allocation results using robust energy-allocation method. $\alpha =\alpha_{1}+\alpha_{2}+\alpha_{3}$. (Target 1: Swerling 1, Target 2: Swerling 2, Target 3: Swerling 3).

**Figure 4 sensors-19-04040-f004:**
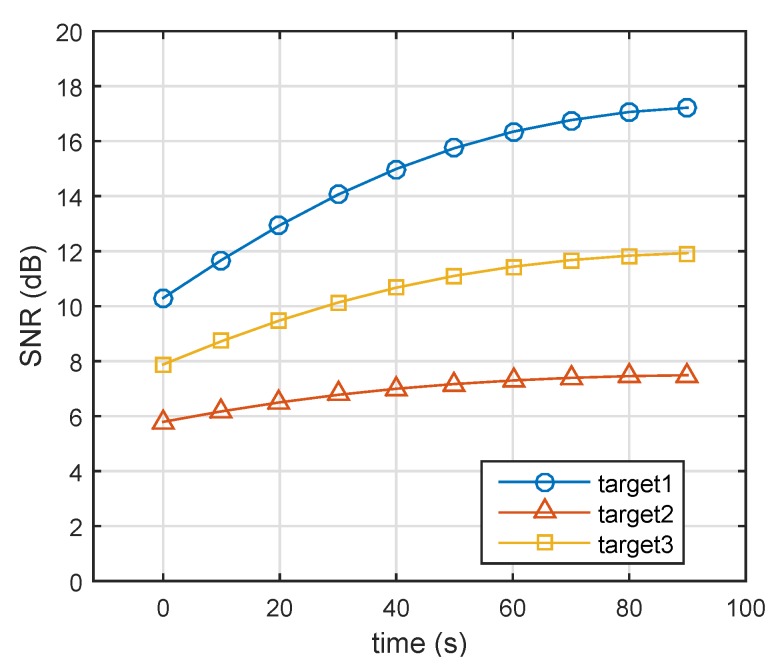
Variations of $SNR_{k}$ using robust energy-allocation method. (Target 1: Swerling 1, Target 2: Swerling 2, Target 3: Swerling 3).

**Figure 5 sensors-19-04040-f005:**
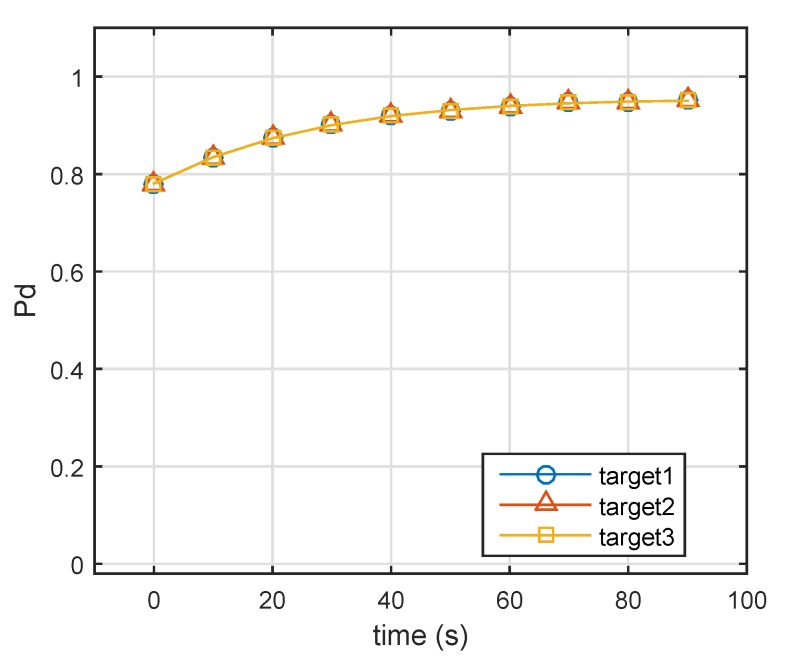
Variation of $P_{d}$ using robust energy-allocation method. (Target 1: Swerling 1, Target 2: Swerling 2, Target 3: Swerling 3).

**Figure 6 sensors-19-04040-f006:**
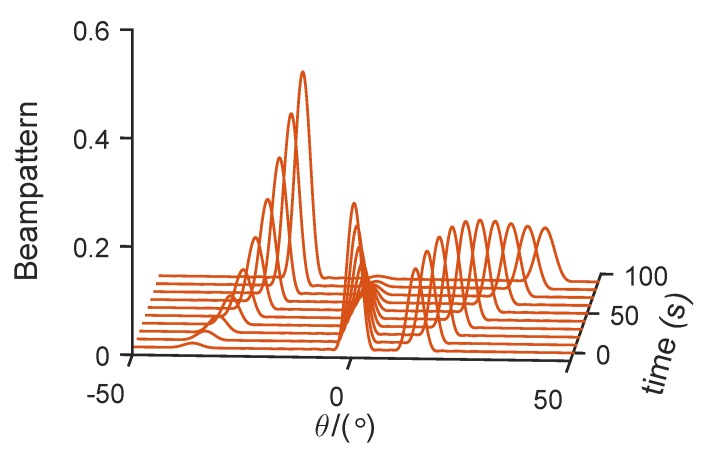
Beam patterns at different times using robust-energy allocation method. (Target 1: Swerling 1, Target 2: Swerling 2, Target 3: Swerling 3).

**Figure 7 sensors-19-04040-f007:**
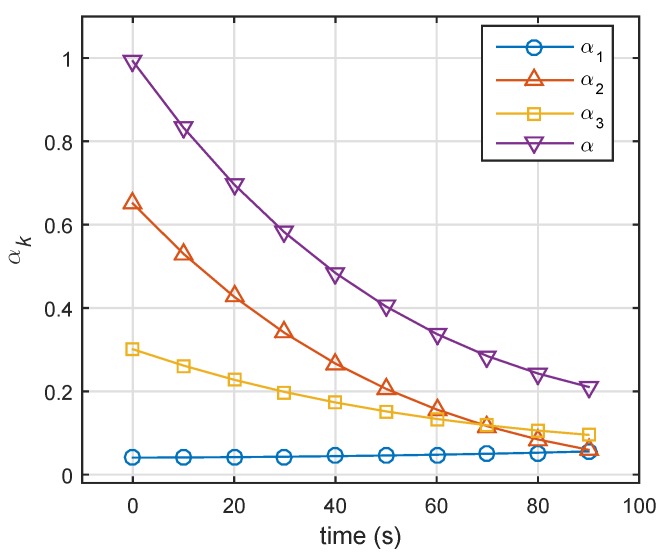
Results of energy allocation using minimum-energy-allocation method (all targets were Swerling 2 targets).

**Figure 8 sensors-19-04040-f008:**
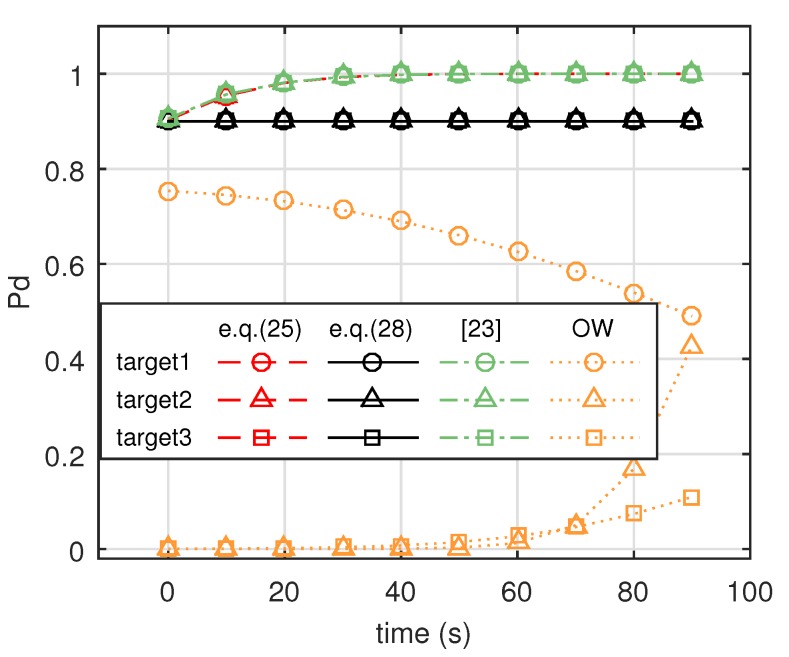
Comparison of $P_{d}$ for three targets with different methods (Equation (Equation 25) denotes robust energy-allocation method, Equation (Equation 28) denotes minimum-energy-allocation method with predefined detection probabilities $\left\{t_{k}=0.9\right\}$, [23] denotes method in [23], and OW denotes orthogonal waveforms. All targets were Swerling 2 targets.)

**Figure 9 sensors-19-04040-f009:**
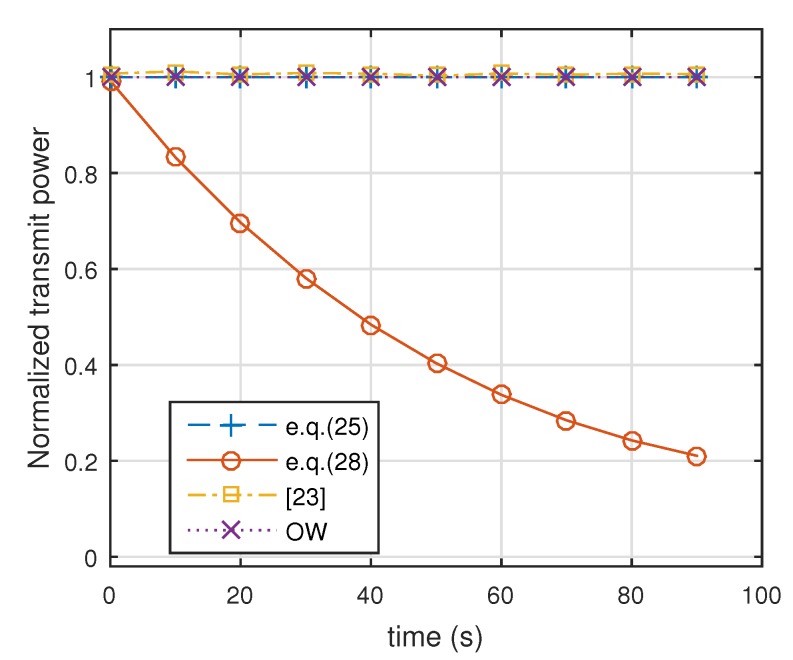
Comparison of transmit power with different methods (Equation (Equation 25) denotes robust energy allocation method, Equation (Equation 28) denotes minimum-energy-allocation method with predefined detection probabilities $\left\{t_{k}=0.9\right\}$, [23] denotes the method in [23], and OW denotes orthogonal waveforms. All targets were Swerling 2 targets.)

**Figure 10 sensors-19-04040-f010:**
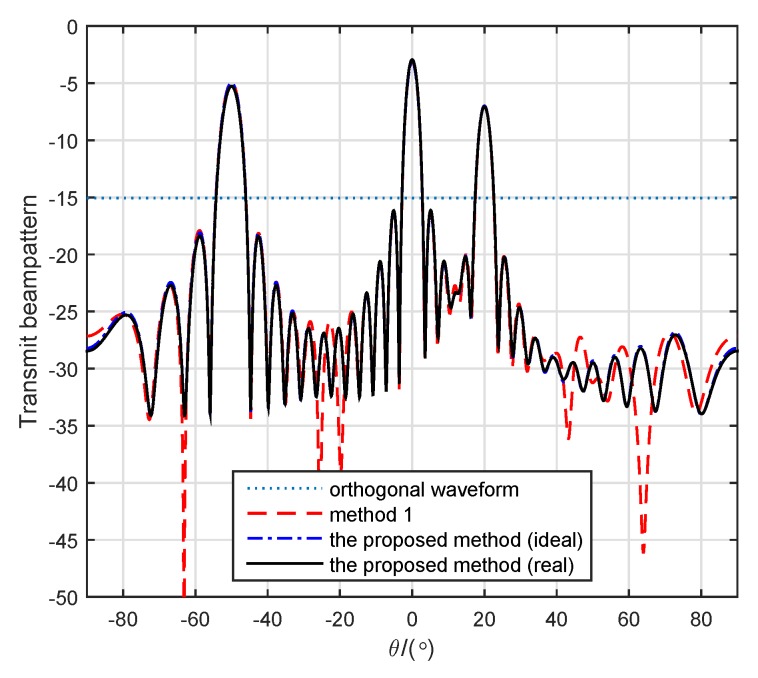
Comparison of beam patterns with different methods.

**Figure 11 sensors-19-04040-f011:**
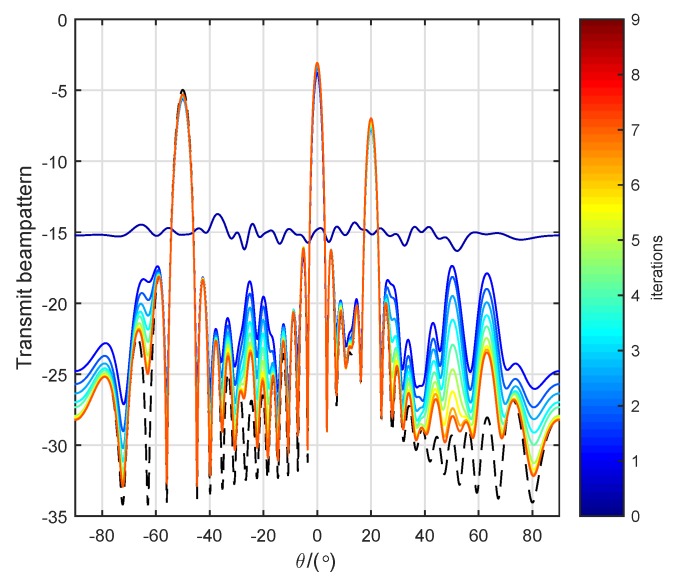
Beam patterns in ten first iterations.

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
