# Peer review of "Constant-Modulus-Waveform Design for Multiple-Target Detection in Colocated MIMO Radar"

_sensors, 2019, doi:10.3390/s19184040_

Round 1

Reviewer 1 Report

General evaluation: This paper addresses the waveform design problem for multi-target detection and tracking tasks.  Generally speaking, the method exploits the weight optimization with ideal orthogonal waveforms to maximize the detection probability of each target (associated with each beam). Then a further constant-modulus regularization method, i.e., cyclic method, is used as to generate the constant-modulus waveforms.  This method is interesting and the technique is sound, thus it merits publication after minor revision.

Comment 1.  The literature review is insufficient.  As there are broad beam and multi-beam designs with the MIMO radar, it is recommended also to cover some works in broad beam design.  Besides, there are some other beampattern design and signal processing within MIMO radar concept, e.g., FDA-MMO radar can form range-angle-dependent beampattern.  Refer to following paper for more details.  Moreover, it is recommended to add more citations on the conventional waveform design methods addressing beampattern design.

(1) Joint range and angle estimation using MIMO radar with frequency diverse array, IEEE Transactions on Signal Processing, 2015.

(2) An adaptive range-angle-Doppler processing approach for FDA-MIMO radar using three-dimensional localization, IEEE Journal of Selected Topics in Signal Processing, 2017

Comment 2.  As target involves time-delay and Doppler shift, how to use the waveform orthogonal condition in this case?

Comment 3:  The expression of fk(ak) should be provided.  It is recommended to add some commentaries on the non-linear property.

Reviewer 2 Report

The authors proposed a two-step waveform design method in collocated MIMO radar to improve the performance of targets detection. There are certain novelty and some good results. However, the English expression needs improvement. It can be accepted if the English expression and the following comments are well revised. 

(1) The authors should introduce the application scenario of the proposed radar system in this paper.

(2) The author needs to further explain how Equation (33) is derived, or to give relevant references, otherwise the context is not coherent.

(3) The legends "eq.(24) and eq.(27)" in Fig. 8 and 9 should be modified to " e.q.(25)" and "e.q.(28)".

(4) The authors should explain the connection between equation (14) and equation (16) after equation (16).

(5) In section two (Signal model), the description of the “θk” variable should be moved forward to where it first appeared.

(6) There are some minor English writing problems.

1) In the third line of the second page, “…more efficient approach is proposed to… ” should be “…more efficient approach was proposed to…”. And in the sixth line, “Using the similar method, paper [9,11] forms desired beampatterns…”, is reference [9] quoted again? “…forms” should be “…form ”.

2) In the sixth line of the second paragraph, “by adjusting the number of beams and their directions and the transmit power of each beam…” should be revised as “by adjusting the number, direction and power of the beams…”.

3) In the first line of the four paragraph, “…obtained by optimize the covariance matrix of …” should be “…obtained by optimizing the covariance matrix of…”.
